# MULTI-DIMENSIONAL EXPLANATION OF REVIEWS

## ABSTRACT

Neural models achieved considerable improvement for many natural language processing tasks, but they offer little transparency, and interpretability comes at a cost. In some domains, automated predictions without justifications have limited applicability. Recently, progress has been made regarding single-aspect sentiment analysis for reviews, where the ambiguity of a justification is minimal. In this context, a justification, or *mask*, consists of (long) word sequences from the input text, which suffice to make the prediction. Existing models cannot handle more than one aspect in one training and induce binary masks that might be ambiguous. In our work, we propose a neural model for predicting multi-aspect sentiments for reviews and generates a *probabilistic multi-dimensional mask* (one per aspect) simultaneously, in an unsupervised and multi-task learning manner. Our evaluation shows that on three datasets, in the beer and hotel domain, our model outperforms strong baselines and generates masks that are: strong feature predictors, meaningful, and interpretable.

## 1 INTRODUCTION

Neural networks have become the standard for many natural language processing tasks. Despite the significant performance gains achieved by these complex models, they offer little *transparency* concerning their inner workings. Thus, they come at the cost of *interpretability* (Jain & Wallace, 2019).

In many domains, automated predictions have a real *impact* on the final decision, such as treatment options in the field of medicine. Therefore, it is important to provide the underlying reasons for such a decision. We claim that integrating interpretability in a (neural) model should supply the reason of the prediction and should yield better performance. However, justifying a prediction might be ambiguous and challenging. Prior work includes various methods that find the justification in an input text — also called rationale or mask of a target variable. The mask is defined as one or multiple pieces of text fragments from the input text.[1] Each should contain words that altogether are short, coherent, and alone sufficient for the prediction as a substitute of the input (Lei et al., 2016).

Many works have been applied to single-aspect sentiment analysis for reviews, where the *ambiguity* of a justification is minimal. In this case, we define an aspect as an attribute of a product or service (Giannakopoulos et al., 2017), such as *Location* or *Cleanliness* for the hotel domain. There are three different methods to generate masks: using reinforcement learning with a trained model (Li et al., 2016b), generating rationales in an unsupervised manner and jointly with the objective function (Lei et al., 2016), or including annotations during training (Bao et al., 2018; Zhang et al., 2016).

However, these models generate justifications that are 1) only tailored for *one* aspect, and 2) expressed as a *hard (binary)* selection of words. A review text reflects opinions about multiple topics a user cares about (Musat et al., 2013). It appears reasonable to analyze multiple aspects with a multi-task learning setting, but a model must be trained as many times as the number of aspects. A hard assignment of words to aspects might lead to ambiguities that are difficult to capture with a *binary mask*: in the text *"The room was large, clean and close to the beach."*, the word *"room"* refers to the aspects *Room*, *Cleanliness* and *Location*. Finally, collecting human-provided rationales at scale is expensive and thus impractical.

In this work, we study interpretable multi-aspect sentiment classification. We describe an architecture for predicting the sentiment of *multiple* aspects while generating a *probabilistic (soft) multi-*

---

[1]In the rest of the paper, we will use the terms mask, justification and rationale interchangeably.

*dimensional mask* (one dimension per aspect) jointly, in an unsupervised and multi-task learning manner. We show that the induced mask is beneficial for identifying simultaneously what parts of the review relate to what aspect, and capturing ambiguities of words belonging to multiple aspects. Thus, the induced mask provides fine-grained interpretability and improves the final performance.

Traditionally interpretability came at a cost of reduced accuracy. In contrast, our evaluation shows that on three datasets, in the beer and hotel domain, our model outperforms strong baselines and generates masks that are: **strong feature predictors, meaningful, and interpretable** compared to attention-based methods and a single-aspect masker. We show that it can be a benefit to 1) guide the model to focus on different parts of the input text, and 2) further improve the sentiment prediction for all aspects. Therefore, interpretabilty does not come at a cost anymore.

The contributions of this work can be summarized as follow:

- We propose a Multi-Aspect Masker (MAM), an end-to-end neural model for multi-aspect sentiment classification that provides fine-grained interpretability in the same training. Given a text review as input, the model generates a probabilistic multi-dimensional mask, with one dimension per aspect. It predicts the sentiments of multiple aspects, and highlights long sequences of words justifying the current rating prediction for each aspect;
- We show that interpretability does not come at a cost: our final model significantly out-performs strong baselines and attention models, both in terms of performance and mask coherence. Furthermore, the level of interpretability is controllable using two regularizers;
- Finally, we release a new dataset for multi-aspect sentiment classification, which contains 140k reviews from TripAdvisor with five aspects, each with its corresponding rating.[2]

## 2 RELATED WORK

### 2.1 INTERPRETABILITY

Developing interpretable models is of considerable interest to the broader research community, even more pronounced with neural models (Kim et al., 2015; Doshi-Velez & Kim, 2017). Many works analyzed and visualized state activation (Karpathy et al., 2015; Li et al., 2016a; Montavon et al., 2018), learned sparse and interpretable word vectors (Faruqui et al., 2015b;a; Herbelot & Vecchi, 2015) or analyzed attention (Clark et al., 2019; Jain & Wallace, 2019). Our work differs from these in terms of what is meant by an explanation. Our system identifies one or multiple short and coherent text fragments that — as a substitute of the input text — **are sufficient for the prediction**.

### 2.2 ATTENTION-BASED MODELS

Attention models (Vaswani et al., 2017; Yang et al., 2016; Lin et al., 2017) have been shown to improve prediction accuracy, visualization, and interpretability. The most popular and widely used attention mechanism is soft attention (Bahdanau et al., 2015) over hard (Luong et al., 2015) and sparse ones (Martins & Astudillo, 2016). According to Jain & Wallace (2019); Serrano & Smith (2019), standard attention modules noisily predict input importance; the weights cannot provide safe and meaningful explanations.

Our model differs in two ways from attention mechanisms: our loss includes two regularizers to favor long word sequences for interpretability; the normalization is not done over the sequence length.

### 2.3 MULTI-ASPECT SENTIMENT CLASSIFICATION

Review multi-aspect sentiment classification is sometimes seen as a sub-problem (Wang et al., 2010; McAuley et al., 2012; Pappas & Popescu-Belis, 2014), by utilizing heuristic-based methods or topic models. Recently, neural models achieved significant improvements with less feature engineering. Yin et al. (2017) built a hierarchical attention model with aspect representations by using a set of manually defined topics. Li et al. (2018) extended this work with user attention and additional features such as overall rating, aspect, and user embeddings. The disadvantage of these methods is their limited interpretability, as they rely on many features in addition to the review text.

---

[2]We will make the code and data available.

## 2.4 RATIONALE-BASED MODELS

The idea of including human rationales during training has been explored in (Zhang et al., 2016; Marshall et al., 2015; Bao et al., 2018). Although they have been shown to be beneficial, they are expensive to collect and might vary across annotators. In our work, no annotation is used.

The work most closely related to ours is Li et al. (2016b) and Lei et al. (2016). Both generate *hard* rationales and address *single-aspect* sentiment classification. Their model must be trained *separately* for each aspect, which leads to ambiguities. Li et al. (2016b) developed a post-training method that removes words from a review text until another trained model changes its prediction. Lei et al. (2016) provides a model that learns an aspect sentiment and its rationale jointly, but hinders the performance and relies on assumptions on the data, such as a small correlation between aspect ratings.

In contrast, our model: 1) supports *multi-aspect* sentiment classification, 2) generates *soft multi-dimensional* masks in a *single* training; 3) the masks provide interpretability and improve the performance significantly.

## 3 METHOD: MULTI-ASPECT MASKER

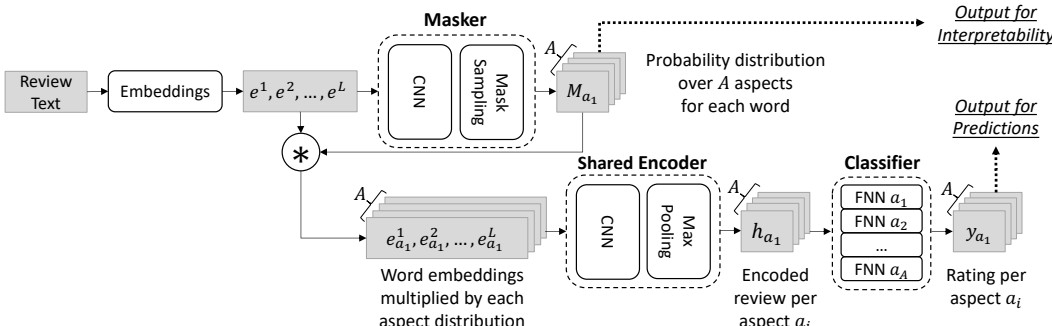

Figure 1: The proposed Multi-Aspect Masker (MAM) model architecture for $A$ aspects.

Let $X$ be a review composed of $L$ words $x^1, x^2, ..., x^L$ and $Y$ the target $A$-dimensional sentiment vector, corresponding to the different rated aspects. Our proposed model, called Multi-Aspect Masker, is composed of three components: 1) a *Masker* module that computes a probability distribution over aspects for each word, resulting in $A + 1$ different masks (including one for *not-aspect*); 2) an *Encoder* that learns a representation of a review conditioned on the induced masks; 3) a *Classifier* that predicts the target variables. The overall model architecture is shown in Figure 1. Our framework generalizes for other tasks, and each neural module is interchangeable with other models.

The *Masker* first computes a hidden representation $h^\ell$ for each word $x^\ell$ in the input sequence, using their word embeddings $e^1, e^2, ..., e^L$. Many sequence models could realize this task, such as recurrent, attention, or convolution neural networks. In our case, we chose a convolutional network because it led to a smaller model, faster training, and empirically, performed similarly to recurrent models. Let $a_i$ denote the $i^{th}$ aspect for $i = 1, ..., A$, and $a_0$ the *not-aspect* case, because many words can be irrelevant to every aspect. We define $M^\ell \in \mathbb{R}^{(A+1)}$, the aspect distribution of the input word $x^\ell$ as:

$$P(\mathbf{M}|X) = \prod_{\ell=1}^{L} P(M^\ell|x^\ell) = \prod_{\ell=1}^{L} \prod_{i=0}^{A} P(m_{a_i}^\ell|x^\ell) \quad (1)$$

Because we have categorical distributions, we cannot directly sample $P(M^\ell|x^\ell)$ and backpropagate the gradient through this discrete generation process. Instead, we model the variable $m_{a_i}^\ell$ using the Straight Through Gumbel Softmax (Jang et al., 2017; Maddison et al., 2017), to approximate sampling from a categorical distribution. We model the parameters of each Gumbel Softmax distribution $M^\ell$ with a single-layer feedforward neural network followed by applying a log softmax, which induces the log-probabilities of the $\ell^{th}$ distribution: $\omega_\ell = \log(\text{softmax}(Wh^\ell + b))$. $W$ and $b$ are shared across all tokens, to have a constant number of parameters with respect to the sequence

length. We control the sharpness of the distributions with the temperature parameter $\tau$. Compared to attention mechanisms, the word importance is a probability distribution over the targets: $\sum_{t=0}^{T} P(m_{a_t}^{\ell}|x^{\ell}) = 1$, instead of a normalization over the sequence length, $\sum_{\ell=1}^{L} P(a^{\ell}|x^{\ell}) = 1$.

Given a soft multi-dimensional mask $\mathbf{M} \in \mathbb{R}^{(A+1) \times L}$, we define each sub-mask $M_{a_i} \in \mathbb{R}^L$ as:

$$M_{a_i} = P(m_{a_i}^1|x^1), P(m_{a_i}^2|x^2), ..., P(m_{a_i}^L|x^L) \tag{2}$$

We weight the word embeddings by their importance towards an aspect $a_i$ with the induced sub-masks, such that $E_{a_i} = M_{a_i} \odot E = P(m_{a_i}^1|x^1) \cdot e_1, \ P(m_{a_i}^2|x^2) \cdot e_2, ..., \ P(m_{a_i}^L|x^L) \cdot e_L$. Thereafter, each modified embedding $E_{a_i}$ is fed into the *Encoder* block. Note that $E_{a_0}$ is ignored because $M_{a_0}$ only serves to absorb probabilities of words that are insignificant to every aspect.[3]

The *Encoder* module includes a convolutional neural network, for the same reasons as earlier, followed by a max-over-time pooling to obtain a fixed-length feature vector. It produces the hidden representation $h_{a_i}$ for each aspect $a_i$. To exploit commonalities and differences among aspects, we share the weights of the encoders for all $E_{a_i}$. Finally, the *Classifier* block contains for each aspect $a_i$ a two-layer feedforward neural networks followed by a softmax layer to predict the sentiment $\hat{y}_{a_i}$.

## 3.1 INTERPRETABLE MASKS

| Attention model | Multi-Aspect Masker |
| --- | --- |
| Trained on $\ell_{sent}$ and no constraint | Trained on $\ell_{sent}$ with $\lambda_p$, $\ell_{sel}$, and $\ell_{cont}$ |

**Aspect Changes:** 30      **Aspect Changes:** 5

Figure 2: Justifications obtained for a hotel review, with an attention model and Multi-Aspect Masker, where the colors represent the aspects: Service, Cleanliness, Value, Location, and Room. Masks lead to mostly long sequences describing clearly each aspect (one switch per aspect), while attention to many short and interleaving sequences (30 changes between aspects), where most relate to noise or multiple aspects. Highlighted words correspond to the highest aspect-attention scores above $1/L$ (i.e., more important than a uniform distribution), and the aspect $a_i$ maximizing $P(m_{a_i}^{\ell}|x^{\ell})$.

The first objective to optimize is the sentiment loss, represented with the cross-entropy between the true aspect sentiment label $y_{a_i}$ and the prediction $\hat{y}_{a_i}$:

$$\ell_{sent} = \sum_{i=1}^{A} \ell_{cross\_entropy}(y_{a_i}, \hat{y}_{a_i}) \tag{3}$$

Training Multi-Aspect Masker to optimize $\ell_{sent}$ will lead to meaningless sub-masks $M_{a_i}$, as the model tends to focus on certain key-words. Consequently, we guide the model to produce long and meaningful sequences of words, as shown in Figure 2. We propose two regularizers: the first controls the number of selected words, and the second favors consecutive words belonging to the

---

[3]if $P(m_{a_0}^{\ell}|x^{\ell}) \approx 1.0$, it implies that $\sum_{i=1}^{A} P(m_{a_i}^{\ell}|x^{\ell}) \approx 0$ and consequently, $e_{a_i}^{\ell} \approx \vec{0}$.

same aspect. For the first term $\ell_{sel}$, we calculate the probability $p_{sel}$ of tagging a word as aspect and then compute the cross-entropy with a parameter $\lambda_p$. The hyper-parameter $\lambda_p$ can be interpreted as the prior on the number of selected words among all aspects, which corresponds to the expectation of Binomial$(p_{sel})$, as the optimizer will try to minimize the difference between $p_{sel}$ and $\lambda_p$.

$$p_{sel} = \frac{1}{L} \sum_{\ell=1}^{L} \left( 1 - P(m_{a_0}^\ell | x^\ell) \right)$$

$$\ell_{sel} = \ell_{binary\_cross\_entropy}(p_{sel}, \lambda_p)$$

(4)

The second regularizer discourages aspect transition between two consecutive words, by minimizing the mean variation of two consecutive aspect distributions. We generalize the formulation in Lei et al. (2016), from a hard binary single-aspect selection, to a soft probabilistic multi-aspect selection.

$$p_{dis} = \frac{1}{L} \sum_{\ell=1}^{L} \left[ \frac{1}{A+1} \sum_{a=0}^{A} | P(m_{a_i}^\ell | x^\ell) - P(m_{a_i}^{\ell-1} | x^{\ell-1}) | \right]$$

$$\ell_{cont} = \ell_{binary\_cross\_entropy}(p_{dis}, 0)$$

(5)

Finally, we train our Multi-Aspect Masker in an end-to-end manner, and optimize the final loss $\ell_{MAM} = \ell_{sent} + \lambda_{sel} \cdot \ell_{sel} + \lambda_{cont} \cdot \ell_{cont}$, where $\lambda_{sel}$ and $\lambda_{cont}$ control the impact of each constraint.

## 4 EXPERIMENTS

In this section, we assess our model on two dimensions: the predictive performance and the quality of the induced mask. We first evaluate Multi-Aspect Masker on the multi-aspect sentiment classification task. In a second experiment, we measure the quality of the induced sub-masks using aspect sentence-level annotations, and an automatic topic model evaluation method.[4]

### 4.1 DATASETS

McAuley et al. (2012) provided 1.5 million beer reviews from BeerAdvocat. Each contains multiple sentences describing various beer aspects: *Appearance*, *Smell*, *Palate*, and *Taste*; users also provided a five-star rating for each aspect. Lei et al. (2016) modified the dataset to suit the requirements of their method.[5] As a consequence, the obtained subset, composed of 280k reviews, *does not reflect the real data distribution*: it contains only the first three aspects, and the sentiment correlation between any pair of aspects is decreased significantly (27.2% on average, instead of 71.8% originally). We denote this version as the *Filtered Beer* dataset, and the original one as the *Full Beer* dataset.

To evaluate the robustness of models across domains, we crawled 140k hotel reviews from TripAdvisor. Each review contains a five-star rating for each aspect: *Service*, *Cleanliness*, *Value*, *Location*, and *Room*. The average correlation between aspects is high (63.0% on average). Compared to beer reviews, hotel reviews are longer, noisier, and less structured, as shown in Appendix A.3.

As in Bao et al. (2018), we binarize the problem: ratings at three and above are labeled as positive and the rest as negative. We further divide the datasets into 80/10/10 for train, development, and test subsets (more details in Appendix A.1).

### 4.2 BASELINES

We compared our Multi-Aspect Masker (MAM) with various baselines. We first used a simple baseline, *Sentiment Majority*, that reports the majority sentiment across aspects, as the sentiment correlation between aspects might be high (see Section 4.1). Because this information is not available at testing, we trained a model to predict the majority sentiment of a review using Wang & Manning (2012). The second baseline we used is a shared encoder followed by $A$ classifiers, that we denote *Emb + Enc + Clf*. This model does not offer any interpretability. We extended it with a

---

[4]The detailed experimental setup is described in Appendix A.2.

[5]For the three first aspects, they trained a simple linear regression model to predict the rating of an aspect given the others and then selected reviews with the largest prediction error.

Table 1: Performance of the best models of each architecture for the ***Filtered Beer*** dataset.

| Interp. | Model | Params | F1 Score | | | |
|---|---|---|---|---|---|---|
| | | | **Macro** | $A_1$ | $A_2$ | $A_3$ |
| None | ⓪ Sentiment Majority | $426k$ | 68.89 | 67.48 | 73.49 | 65.69 |
| | ① $\text{Emb}_{200} + \text{Enc}_{\text{CNN}} + \text{Clf}$ | $173k$ | 78.23 | 78.38 | 80.86 | 75.47 |
| Coarse-grained | ② $\text{Emb}_{200} + \text{Enc}_{\text{CNN}} + \text{A}_{\text{Shared}} + \text{Clf}$ | $196k$ | 78.19 | 77.43 | 80.96 | 76.16 |
| | ③ $\text{Emb}_{200} + \text{Enc}_{\text{LSTM}} + \text{A}_{\text{Shared}} + \text{Clf}$ | $186k$ | 78.16 | 75.88 | 81.25 | 77.36 |
| Fine-grained | ④ NB-SVM (Wang & Manning, 2012) | $3 \cdot 426k$ | 74.60 | 73.50 | 77.32 | 72.99 |
| | ⑤ SAM (Lei et al., 2016) | $3 \cdot 644k$ | 77.06 | 77.36 | 78.99 | 74.83 |
| | ⑥ $\text{Emb}_{200} + \text{Enc}_{\text{LSTM}} + \text{A}^{\text{Sparse}}_{\text{Aspect-wise}} + \text{Clf}$ | $458k$ | 78.82 | 77.35 | 81.65 | 77.47 |
| | ⑦ $\text{Emb}_{200} + \text{Enc}_{\text{LSTM}} + \text{A}_{\text{Aspect-wise}} + \text{Clf}$ | $458k$ | 78.96 | 78.54 | 81.56 | 76.79 |
| | ⑧ $\text{Emb}_{200} + \text{Masker} + \text{Enc}_{\text{CNN}} + \text{Clf}$ (Ours) | $274k$ | 79.32 | 78.58 | 81.71 | 77.66 |
| | ⑨ $\textbf{Emb}_{200+3} + \textbf{Enc}_{\textbf{CNN}} + \textbf{Clf}$ **(Ours)** | $175k$ | **79.66** | **78.74** | **82.02** | **78.22** |

shared attention mechanism (Bahdanau et al., 2015) after the encoder, noted $\text{A}_{\text{Shared}}$, that provides a *coarse-grained interpretability*: for all aspects, the network focuses on the same words in the input.

Our final goal is to achieve the best performance and provide *fine-grained interpretability*: to visualize what sequences of words a model focuses on and to predict the aspect sentiment predictions. To this end, we included other baselines: two trained *separately* for each aspect and two trained with a *multi-aspect* sentiment loss. We employed for the first ones: the well-known NB-SVM of Wang & Manning (2012) for sentiment analysis tasks, and the Single Aspect-Mask (SAM) model from Lei et al. (2016), each trained separately for each aspect. The two last methods are composed of a separate encoder, attention mechanism, and classifier for each aspect. We utilized two types of attention mechanism: additive (Bahdanau et al., 2015), and sparse (Martins & Astudillo, 2016). We call each variant Multi Aspect-Attentions (MAA) and Multi Aspect-Sparse-Attentions (MASA). Diagrams for the baselines can be found in Appendix A.5.

### 4.3 MULTI-ASPECT SENTIMENT CLASSIFICATION

In this section, we enquire whether fine-grained interpretability can become a benefit rather than a cost. We group the models and baselines in three different levels of interpretability:

- *None*: we cannot identify what parts of the review are important for the prediction;
- *Coarse-grained*: we can identify what parts of the reviews were important to predict **all** aspect sentiments, without knowing what part corresponds to what aspect;
- *Fine-grained*: for each aspect, we can identify what parts are used to predict its sentiment.

#### 4.3.1 BEER REVIEWS

Overall F1 scores (macro and for each aspect $A_i$) for the controlled-environment *Filtered Beer* (where there are assumptions on the data distribution) and the real-world *Full Beer* dataset are shown in Table 1 and Table 2.

We find that our Multi-Aspect Masker (MAM) model ⑧, with 1.7 to 2.1 times fewer parameters than aspect-wise attention models (⑥ + ⑦), performs better on average than all other baselines on both datasets, and provides fine-grained interpretability. For the synthetic *Filtered Beer* dataset, MAM achieves a significant improvement of at least 0.36 macro F1 score, and 0.05 for the *Full Beer* one.

To demonstrate that the induced sub-masks $M_{a_1}, ..., M_{a_A}$ are 1) meaningful for other models to improve final predictions, and 2) bring fine-grained interpretability, we extracted and concatenated the masks to the word embeddings, resulting in contextualized embeddings (Peters et al., 2018).

Table 2: Performance of the best models of each architecture for the **_Full Beer_** dataset.

| Interp. | Model | Params | F1 Score | | | | |
|---|---|---|---|---|---|---|---|
| | | | **Macro** | $A_1$ | $A_2$ | $A_3$ | $A_4$ |
| None | ⓪ Sentiment Majority | $560k$ | 73.01 | 71.83 | 75.65 | 71.26 | 73.31 |
| | ① $\text{Emb}_{200} + \text{Enc}_{\text{CNN}} + \text{Clf}$ | $188k$ | 76.45 | 71.44 | 78.64 | 74.88 | 80.83 |
| Coarse-grained | ② $\text{Emb}_{200} + \text{Enc}_{\text{CNN}} + \text{A}_{\text{Shared}} + \text{Clf}$ | $226k$ | 77.06 | 73.44 | 78.68 | 75.79 | 80.32 |
| | ③ $\text{Emb}_{200} + \text{Enc}_{\text{LSTM}} + \text{A}_{\text{Shared}} + \text{Clf}$ | $219k$ | 78.03 | 74.25 | 79.53 | 75.76 | 82.57 |
| Fine-grained | ④ NB-SVM (Wang & Manning, 2012) | $4 \cdot 560k$ | 72.11 | 72.03 | 74.95 | 68.11 | 73.35 |
| | ⑤ SAM (Lei et al., 2016) | $4 \cdot 644k$ | 76.62 | 72.93 | 77.94 | 75.70 | 79.91 |
| | ⑥ $\text{Emb}_{200} + \text{Enc}_{\text{LSTM}} + \text{A}_{\text{Aspect-wise}}^{\text{Sparse}} + \text{Clf}$ | $611k$ | 77.62 | 72.75 | 79.62 | 75.81 | 82.28 |
| | ⑦ $\text{Emb}_{200} + \text{Enc}_{\text{LSTM}} + \text{A}_{\text{Aspect-wise}} + \text{Clf}$ | $611k$ | 78.50 | 74.58 | 79.84 | 77.06 | 82.53 |
| | ⑧ $\text{Emb}_{200} + \text{Masker} + \text{Enc}_{\text{CNN}} + \text{Clf}$ (Ours) | $289k$ | 78.55 | 74.87 | 79.93 | 77.39 | 82.02 |
| | ⑨ $\mathbf{Emb_{200+4} + Enc_{CNN} + Clf}$ **(Ours)** | $191k$ | **78.94** | **75.02** | **80.17** | **77.86** | **82.71** |

We trained a simple *Encoder-Classifier* with the contextualized embeddings ⑨, which has approximately $1.5$ times fewer parameters than MAM. We achieved a macro F1 score absolute improvement of $0.34$ compared to MAM, and $1.43$ compared to the non-contextualized variant for the *Filtered Beer* dataset; for the *Full Beer* one, the performance increases by $0.39$ and $2.49$ respectively. Similarly, each individual aspect $A_i$ F1 score of MAM is improved to a similar extent.

We provide in Appendix A.3.1 and A.3.2 visualizations of reviews with the computed sub-masks $M_{a_1}, ..., M_{a_A}$ and attentions by different models. Not only do sub-masks enable the reach of higher performance; they better capture parts of reviews related to each aspect compared to other methods.

Both NB-SVM ④ and SAM ⑤, offering fine-grained interpretability and trained separately for each aspect, significantly underperform compared to the *Encoder-Classifier* ①. This result is expected: the goal of SAM is to provide rationales at the price of performance (Lei et al., 2016), and NB-SVM might not perform well because of its simplicity. Shared attention models (② + ③) perform similarly to the *Encoder-Classifier* ①, but provide only coarse-grained interpretability.

However, in the *Full Beer* dataset, SAM ⑤ obtains better results than the *Encoder-Classifier* baseline ① and NB-SVM ④, which is outperformed by all other models. It might be counterintuitive that SAM performs better, but we claim that its behavior comes from the high correlation between aspects: SAM selects words that should belong to aspect $a_i$ to predict the sentiment of aspect $a_j$ ($a_i \neq a_j$). Moreover, in Section 4.5, we show that a single-aspect mask from SAM cannot be employed for interpretability.

Finally, *Sentiment Majority* ⓪ is outperformed by a large margin by all other models in the *Filtered Beer* dataset, because of the low sentiment correlation between aspects. However, in the realistic dataset *Full Beer*, *Sentiment Majority* obtains higher score and performs better than NB-SVM ④. These results emphasize the ease of the *Filtered Beer* dataset compared to the *Full Beer* one, because of the assumptions not holding in the real data distribution.

### 4.3.2 MODEL ROBUSTNESS - HOTEL REVIEWS

On the *Hotel* dataset, the learned mask M from Multi-Aspect Masker ⑧ is again meaningful, by increasing the performance and adding interpretability. The *Encoder-Classifier* with contextualized embeddings ⑨ outperforms all other models significantly, with an absolute macro F1 score improvement of $0.49$. Moreover, it achieves the best individual F1 score for each aspect $A_1, ..., A_5$.

Visualizations of reviews, with masks and attentions, are available in Appendix A.3.3. The interpretability comes from the long sequences that MAM identifies, unlike attention models. In comparison, SAM ⑤ lacks coverage and suffers from ambiguity due to the high correlation between aspects. We observe that Multi-Aspect Masker ⑧ performs slightly worse than aspect-wise atten-

Table 3: Performance of the best models of each architecture for the ***Hotel*** dataset.

| Interp. | Model | Params | F1 Score | | | | | |
|---------|-------|--------|-------|-------|-------|-------|-------|-------|
| | | | **Macro** | $A_1$ | $A_2$ | $A_3$ | $A_4$ | $A_5$ |
| None | ⓪ Sentiment Majority | $309k$ | 85.91 | 89.98 | 90.70 | 92.12 | 65.09 | 91.67 |
| | ① $Emb_{300} + Enc_{CNN} + Clf$ | $263k$ | 90.30 | 92.91 | 93.55 | 94.12 | 76.65 | 94.29 |
| Coarse-grained | ② $Emb_{300} + Enc_{CNN} + A_{Shared} + Clf$ | $301k$ | 90.12 | 92.73 | 93.55 | 93.76 | 76.40 | 94.17 |
| | ③ $Emb_{300} + Enc_{LSTM} + A_{Shared} + Clf$ | $270k$ | 88.22 | 91.13 | 92.19 | 93.33 | 71.40 | 93.06 |
| Fine-grained | ④ NB-SVM (Wang & Manning, 2012) | $5 \cdot 309k$ | 87.17 | 90.04 | 90.77 | 92.30 | 71.27 | 91.46 |
| | ⑤ SAM (Lei et al., 2016) | $5 \cdot 824k$ | 87.52 | 91.48 | 91.45 | 92.04 | 70.80 | 91.85 |
| | ⑥ $Emb_{200} + Enc_{LSTM} + A_{Aspect\text{-}wise}^{Sparse} + Clf$ | $1010k$ | 90.23 | 93.11 | 93.32 | 93.58 | 77.21 | 93.92 |
| | ⑦ $Emb_{300} + Enc_{LSTM} + A_{Aspect\text{-}wise} + Clf$ | $1010k$ | 90.21 | 92.84 | 93.34 | 93.78 | 76.87 | 94.21 |
| | ⑧ $Emb_{300} + Masker + Enc_{CNN} + Clf$ (Ours) | $404k$ | 89.94 | 92.84 | 92.95 | 93.91 | 76.27 | 93.71 |
| | ⑨ **$Emb_{300+5} + Enc_{CNN} + Clf$ (Ours)** | $267k$ | **90.79** | **93.38** | **93.82** | **94.55** | **77.47** | **94.71** |

Table 4: Precision of selected words for each aspect. Percentage of words corresponds to the ratio of the number of highlighted words to the full review. All models are trained on *Filtered Beer*.

| Interp. | Model | *Smell* | | *Palate* | | *Appearance* | |
|---------|-------|---------|---------|----------|---------|--------------|---------|
| | | **Prec.** | **%Words** | **Prec.** | **%Words** | **Prec.** | **%Words** |
| Fine-grained | SVM[*] (Lei et al., 2016) | 21.6 | 7% | 24.9 | 7% | 38.3 | 13% |
| | SAA[*] | 88.4 | 7% | 65.3 | 7% | 80.6 | 13% |
| | SAM[*] (Lei et al., 2016) | 95.1 | 7% | 80.2 | 7% | 96.3 | 14% |
| | MASA | 87.0 | 4% | 42.8 | 5% | 74.5 | 4% |
| | MAA | 51.3 | 7% | 32.9 | 7% | 44.9 | 14% |
| | **MAM (Ours)** | **96.6** | 7% | **81.7** | 7% | **96.7** | 14% |

[*] The model has been trained separately for each aspect.

tion models (⑦ + ⑧), with 2.5 times fewer parameters. We emphasize that using the induced masks in the *Encoder-Classifier* ⑨ already achieves the best performance.

The Single Aspect-Mask ⑤ obtains the lowest relative macro F1 score of the three datasets: a difference of $-3.27$; $-2.6$ and $-2.32$ for the *Filtered Beer* and *Full Beer* dataset respectively. This proves that the model is not meant to provide rationales and increase the performance simultaneously.

Finally, we show that learning soft multi-dimensional masks along training objectives achieves strong predictive results, and using these to create contextualized word embeddings and train a baseline model with, provides the best performance across the three datasets.

## 4.4 MASK INTERPRETABILITY

In these experiments, we verify that Multi-Aspect Masker generates induced masks $M_{a_1}, ..., M_{a_A}$ that, in addition to improving performance, are meaningful and can be interpreted by humans.

### 4.4.1 MASK PRECISION

Evaluating justifications that have short and coherent pieces of text is challenging because there is no gold standard provided with reviews. McAuley et al. (2012) provided 994 beer reviews with aspect sentence-level annotations, although our model computes masks at a finer level. Each sentence of the

Table 5: Average Topic Coherence (NPMI) across different top-$N$ words for each dataset. Each aspect $a_i$ is considered as a topic and the masks (or attentions) are used to compute $P(w|a_i)$.

| | **Model** | $N=5$ | $N=10$ | $N=15$ | $N=20$ | $N=25$ | $N=30$ | Mean[†] |
|---|---|---|---|---|---|---|---|---|
| Filtered Beer | SAM[*] (Lei et al., 2016) | **0.123** | **0.149** | 0.134 | 0.169 | 0.219 | 0.248 | 0.174 |
| | MASA | 0.024 | 0.059 | 0.159 | 0.200 | 0.271 | 0.325 | 0.173 |
| | MAA | 0.072 | 0.103 | 0.141 | 0.208 | 0.259 | 0.325 | 0.185 |
| | MAM (Ours) | 0.042 | 0.114 | **0.171** | **0.216** | **0.276** | **0.329** | **0.192** |
| Full Beer | SAM[*] (Lei et al., 2016) | 0.046 | 0.120 | 0.129 | 0.243 | 0.308 | 0.396 | 0.207 |
| | MASA | 0.020 | 0.082 | 0.130 | 0.168 | 0.234 | 0.263 | 0.150 |
| | MAA | 0.064 | **0.189** | 0.255 | 0.273 | 0.332 | 0.401 | 0.252 |
| | MAM (Ours) | **0.083** | 0.187 | **0.264** | **0.348** | **0.410** | **0.477** | **0.295** |
| Hotel | SAM[*] (Lei et al., 2016) | 0.041 | 0.103 | 0.152 | 0.180 | 0.233 | 0.281 | 0.165 |
| | MASA | 0.043 | 0.127 | 0.166 | 0.295 | 0.323 | 0.458 | 0.235 |
| | MAA | 0.128 | 0.218 | **0.352** | 0.415 | 0.494 | 0.553 | 0.360 |
| | MAM (Ours) | **0.134** | **0.251** | 0.349 | **0.496** | **0.641** | **0.724** | **0.432** |

[*] The model has been trained separately for each aspect.
[†] Metric that correlates best with human judgment (Lau & Baldwin, 2016).

dataset is annotated with one aspect label, indicating what aspect that sentence covers. We evaluate the precision of words highlighted by each model. For both, ours and Lei et al. (2016), we used trained models on beer reviews and extracted a similar number of selected words.

We show that the generated sub-masks $M_{a_1}, M_{a_2}, M_{a_3}$ obtained with Multi-Aspect Masker (MAM) correlate best with human judgment. Table 4 presents the precision of the masks and attentions computed on sentence-level aspect annotations. We reported results of the models in Lei et al. (2016): SVM, the Single Aspect-Attention (SAA) and Single Aspect-Mask (SAM) — trained *separately* for each aspect because they find *hard* justifications for a single aspect. In comparison to SAM, our MAM model obtains significant higher precisions with an average of $+1.13$. Interestingly, SVM and attention models perform poorly compared with mask models: especially MASA that focuses only on a couple of words due to the sparseness of the attention (examples in Appendix A.3.1).

## 4.5 MASK COHERENCE

In addition to evaluating masks with human annotations, we computed their semantic interpretability for each dataset. According to Aletras & Stevenson (2013); Lau et al. (2014), NPMI (Bouma, 2009) is a good metric for qualitative evaluation of topics, because it matches human judgment most closely. However, the top-$N$ topic words, used for evaluation, are often selected arbitrarily. To alleviate this problem, we followed Lau & Baldwin (2016): we computed the topic coherence over several cardinalities $N$, and report all the results, as well as the average; the authors claim the mean leads to a more stable and robust evaluation. More details are available in Appendix A.4.

We show that generated masks by MAM obtains the highest mean NPMI and, on average, superior results in all datasets (17 out of 21 cases), while only needing a single training. Results are shown in Table 5. For the *Hotel* and *Full Beer* datasets, where reviews reflect the real data distribution, our model significantly outperforms SAM and attention models for $N \geq 20$. For smaller $N$, MAM obtains higher scores in four out of six cases, and for these two, the difference is only below 0.003.

For the controlled-environment *Filtered Beer* dataset, MAM still performs better for $N \geq 15$, although the differences are smaller, and is beat by SAM for $N \leq 10$. However, SAM obtains poor results in all other cases of all datasets and must be trained as many times as the number of aspects.

We show the top words for each aspect and a human evaluation in Appendix A.4. Generally, our model finds better sets of words among the three datasets compared with other methods.

## 5 CONCLUSION

In this work, we propose Multi-Aspect Masker, an end-to-end neural network architecture to perform multi-aspect sentiment classification for reviews. Our model predicts aspect sentiments while generating a *probabilistic (soft) multi-dimensional mask* (one dimension per aspect) simultaneously, in an unsupervised and multi-task learning manner. We showed that the induced mask is beneficial to guide the model to focus on different parts of the input text and to further improve the sentiment prediction for all aspects. Our evaluation shows that on three datasets, in the beer and hotel domain, our model outperforms strong baselines and generates masks that are: **strong feature predictors, meaningful, and interpretable** compared to attention-based methods and a single-aspect masker.

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

# A   APPENDIX

## A.1   DESCRIPTIVE STATISTICS OF THE DATASETS

Table 6: Statistics of the multi-aspect review datasets. *Full Beer* and *Hotel* represent real-world beer and hotel reviews respectively. *Filtered Beer* contains a subset of beer reviews with assumptions, leading to a more straightforward and unrealistic dataset.

| Dataset | Filtered Beer | Full Beer | Hotel |
|---|---|---|---|
| Number of reviews | $280,000$ | $1,586,259$ | $140,000$ |
| Average word-length of review | $157.5 \pm 84.3$ | $147.1 \pm 79.7$ | $188.3 \pm 50.0$ |
| Average sentence-length of review | $11.0 \pm 5.7$ | $10.3 \pm 5.4$ | $10.4 \pm 4.4$ |
| Number of aspects | 3 | 4 | 5 |
| Average ratio of $\oplus$ over $\ominus$ reviews per aspect | 3.29 | 12.89 | 1.02 |
| Average correlation between aspects | **27.2**% | **71.8**% | **63.0**% |
| Max correlation between two aspects | **29.8**% | **73.4**% | **86.5**% |

## A.2   EXPERIMENTAL DETAILS

For each model, the review encoder was either a bi-directional single-layer forward recurrent neural network using Long Short-Term Memory (Hochreiter & Schmidhuber, 1997) with 64 hidden units or the multi-channel text convolutional neural network, similar to Kim et al. (2015), with 3, 5, 7 width filters and 50 feature maps per filter. Each aspect classifier is a two-layer feedforward neural network with ReLU activation function (Nair & Hinton, 2010). We used the 200-dimensional pre-trained word embeddings of Lei et al. (2016) for beer reviews. For the hotel domain, we trained word2vec (Mikolov et al., 2013) on a large collection of hotel reviews with an embedding size of 300.

We used dropout (Srivastava et al., 2014) of 0.1, clipped the gradient norm at 1.0 if higher, added L2-norm regularizer with a regularization factor of $10^{-6}$ and trained using early stopping with a patience of three iterations. We used Adam (Kingma & Ba, 2015) for training with a learning rate of 0.001, $\beta_1 = 0.9$, and $\beta_2 = 0.999$. The temperature $\tau$ for Gumbel-Softmax distributions was fixed at 0.8. The two regularizer terms and the prior of our model are $\lambda_{sel} = 0.03$, $\lambda_{cont} = 0.04$, and $\lambda_p = 0.11$ for the *Filtered Beer* dataset; $\lambda_{sel} = 0.03$, $\lambda_{cont} = 0.03$, and $\lambda_p = 0.15$ for the *Full Beer* dataseet; and $\lambda_{sel} = 0.02$, $\lambda_{cont} = 0.02$ and $\lambda_p = 0.10$ for the *Hotel* dataset. We ran all experiments for a maximum of 50 epochs with a batch-size of 256 and a Titan X GPU. For the model of Lei et al. (2016), we reused the code from the authors.

## A.3   VISUALIZATION OF THE MULTI-DIMENSIONAL FACETS OF REVIEWS

We randomly sampled reviews from each dataset and computed the masks and attentions of four models: our Multi-Aspect Masker model (MAM), the Single Aspect-Mask method (SAM) of Lei et al. (2016) and two attention models with additive and sparse attention, called Multi Aspect-Attentions (MAA) and Multi Aspect-Sparse-Attentions (MASA) respectively (more details in Section 4.2). Each color represents an aspect and the shade its confidence. All models generate soft attentions or masks besides SAM, which does hard masking. Samples for the *Filtered Beer*, *Full Beer* and *Hotel* dataset are shown below.

### A.3.1 Filtered Beer Dataset

Appearance Smell Palate

**Multi Aspect-Masks (Ours)**

a : ruby red brown in color . fluffy off white single - finger head settles down to a thin cap . coating thin lacing all over the sides on the glass . s : some faint burnt , sweet malt smells , but little else and very faint . t : taste is very solid for a brown . malts and some sweetness . maybe some toffee , biscuit and burnt flavors too . m : decent carbonation is followed by a medium bodied feel . flavor coats the tongue for a very satisfying and lasting finish . d : an easy drinker , as a good brown should be . my wife is a big brown fan , so i 'll definitely be grabbing this one for her again . a solid beer for any time of the year . served : in a standard pint glass .

Appearance Smell Palate

Single Aspect-Mask (Lei et al., 2016)

a : ruby red brown in color . fluffy off white single - finger head settles down to a thin cap . coating thin lacing all over the sides on the glass . s : some faint burnt , sweet malt smells , but little else and very faint . t : taste is very solid for a brown . malts and some sweetness . maybe some toffee , biscuit and burnt flavors too . m : decent carbonation is followed by a medium bodied feel . flavor coats the tongue for a very satisfying and lasting finish . d : an easy drinker , as a good brown should be . my wife is a big brown fan , so i 'll definitely be grabbing this one for her again . a solid beer for any time of the year . served : in a standard pint glass .

Appearance Smell Palate

Multi Aspect-Attentions

a : ruby red brown in color . fluffy off white single - finger head settles down to a thin cap . coating thin lacing all over the sides on the glass . s : some faint burnt , sweet malt smells , but little else and very faint . t : taste is very solid for a brown . malts and some sweetness . maybe some toffee , biscuit and burnt flavors too . m : decent carbonation is followed by a medium bodied feel . flavor coats the tongue for a very satisfying and lasting finish . d : an easy drinker , as a good brown should be . my wife is a big brown fan , so i 'll definitely be grabbing this one for her again . a solid beer for any time of the year . served : in a standard pint glass .

Appearance Smell Palate

Multi Aspect-Sparse-Attentions

a : ruby red brown in color . fluffy off white single - finger head settles down to a thin cap . coating thin lacing all over the sides on the glass . s : some faint burnt , sweet malt smells , but little else and very faint . t : taste is very solid for a brown . malts and some sweetness . maybe some toffee , biscuit and burnt flavors too . m : decent carbonation is followed by a medium bodied feel . flavor coats the tongue for a very satisfying and lasting finish . d : an easy drinker , as a good brown should be . my wife is a big brown fan , so i 'll definitely be grabbing this one for her again . a solid beer for any time of the year . served : in a standard pint glass .

Figure 3: Our model MAM highlights all the words corresponding to aspects. SAM only highlights the most crucial information, but some words are missing out, and one is ambiguous. MAA and MASA fail to identify most of the words related to the aspect *Appearance*, and only a few words have high confidence, resulting in noisy labeling. Additionally, MAA considers words belonging to the aspect *Taste* whereas the *Filtered Beer* dataset does not include it in the aspect set.

Appearance Smell Palate

**Multi Aspect-Masks (Ours)**

a- crystal clear gold , taunt fluffy three finger white head that holds it own very well , when it falls it falls to a 1/2 " ring , full white lace on glass s- clean , crisp , floral , pine , citric lemon t- crisp biscuit malt up front , hops all the way through , grassy , lemon , tart yeast at finish , hop bitterness through finish m- dry , bubbly coarse , high carbonation , light bodied , hops leave impression on palette . d- nice hop bitterness , good flavor , sessionable , recommended , good brew

Appearance Smell Palate

Single Aspect-Mask (Lei et al., 2016)

a- crystal clear gold , taunt fluffy three finger white head , that holds it own very well , when it falls it falls to a 1/2 " ring , full white lace on glass s- clean , crisp , floral , pine , citric lemon t- crisp biscuit malt up front , hops all the way through , grassy , lemon , tart yeast at finish , hop bitterness through finish m- dry , bubbly coarse , high carbonation , light bodied , hops leave impression on palette . d- nice hop bitterness , good flavor , sessionable , recommended , good brew

Appearance Smell Palate

Multi Aspect-Attentions

a- crystal clear gold , taunt fluffy three finger white head that holds it own very well , when it falls it falls to a 1/2 " ring , full white lace on glass s- clean , crisp , floral , pine , citric lemon t- crisp biscuit malt up front , hops all the way through , grassy , lemon , tart yeast at finish , hop bitterness through finish m- dry , bubbly coarse , high carbonation , light bodied , hops leave impression on palette . d- nice hop bitterness , good flavor , sessionable , recommended , good brew

Appearance Smell Palate

Multi Aspect-Sparse-Attentions

a- crystal clear gold , taunt fluffy three finger white head that holds it own very well , when it falls it falls to a 1/2 " ring , full white lace on glass s- clean , crisp , floral , pine , citric lemon t- crisp biscuit malt up front , hops all the way through , grassy , lemon , tart yeast at finish , hop bitterness through finish m- dry , bubbly coarse , high carbonation , light bodied , hops leave impression on palette . d- nice hop bitterness , good flavor , sessionable , recommended , good brew

Figure 4: MAM finds the exact parts corresponding to the aspect *Appearance* and *Palate* while covering most of the aspect *Smell*. SAM identifies key-information without any ambiguity, but lacks coverage. MAA highlights confidently nearly all the words while having some noise for the aspect *Appearance*. MASA selects confidently only most predictive words.

## A.3.2 FULL BEER DATASET

Appearance Smell Palate Taste

**Multi Aspect-Masks (Ours)**

sa 's harvest pumpkin ale 2011 . had this last year , loved it , and bought 6 harvest packs and saved the pumpkins and the dunkel 's ... not too sure why sa dropped the dunkel , i think it would make a great standard to them . pours a dark brown with a 1 " bone white head , that settles down to a thin lace across the top of the brew . smells of the typical pumpkin pie spice , along with a good squash note . tastes just like last years , very subtle , nothing over the top . a damn good pumpkin ale that is worth seeking out . when i mean everything is subtle i mean everything . nothing is overdone in this pumpkin ale , and is a great representation of the original style . mouthfeel is somewhat thick , with a pleasant coating feel . overall , i loved it last year , and i love it this year . do n't get me wrong , its no pumpking , but this is a damn fine pumpkin ale that could hold its own any day among all the others . i would rate this as my 4th favorite pumpkin ale to date . i 'm not sure why the bros rated it so low , but do n't take their opinion , make your own !

Appearance Smell Palate Taste

Single Aspect-Mask (Lei et al., 2016)

sa 's harvest pumpkin ale 2011 . had this last year , loved it , and bought 6 harvest packs and saved the pumpkins and the dunkel 's ... not too sure why sa dropped the dunkel , i think it would make a great standard to them . pours a dark brown with a 1 " bone white head , that settles down to a thin lace across the top of the brew . smells of the typical pumpkin pie spice , along with a good squash note . tastes just like last years , very subtle , nothing over the top . a damn good pumpkin ale that is worth seeking out . when i mean everything is subtle i mean everything . nothing is overdone in this pumpkin ale , and is a great representation of the original style . mouthfeel is somewhat thick , with a pleasant coating feel . overall , i loved it last year , and i love it this year . do n't get me wrong , its no pumpking , but this is a damn fine pumpkin ale that could hold its own any day among all the others . i would rate this as my 4th favorite pumpkin ale to date . i 'm not sure why the bros rated it so low , but do n't take their opinion , make your own !

Appearance Smell Palate Taste

Multi Aspect-Attentions

sa 's harvest pumpkin ale 2011 . had this last year , loved it , and bought 6 harvest packs and saved the pumpkins and the dunkel 's ... not too sure why sa dropped the dunkel , i think it would make a great standard to them . pours a dark brown with a 1 " bone white head , that settles down to a thin lace across the top of the brew . smells of the typical pumpkin pie spice , along with a good squash note . tastes just like last years , very subtle , nothing over the top . a damn good pumpkin ale that is worth seeking out . when i mean everything is subtle i mean everything . nothing is overdone in this pumpkin ale , and is a great representation of the original style . mouthfeel is somewhat thick , with a pleasant coating feel . overall , i loved it last year , and i love it this year . do n't get me wrong , its no pumpking , but this is a damn fine pumpkin ale that could hold its own any day among all the others . i would rate this as my 4th favorite pumpkin ale to date . i 'm not sure why the bros rated it so low , but do n't take their opinion , make your own !

Appearance Smell Palate Taste

Multi Aspect-Sparse-Attentions

sa 's harvest pumpkin ale 2011 . had this last year , loved it , and bought 6 harvest packs and saved the pumpkins and the dunkel 's ... not too sure why sa dropped the dunkel , i think it would make a great standard to them . pours a dark brown with a 1 " bone white head , that settles down to a thin lace across the top of the brew . smells of the typical pumpkin pie spice , along with a good squash note . tastes just like last years , very subtle , nothing over the top . a damn good pumpkin ale that is worth seeking out . when i mean everything is subtle i mean everything . nothing is overdone in this pumpkin ale , and is a great representation of the original style . mouthfeel is somewhat thick , with a pleasant coating feel . overall , i loved it last year , and i love it this year . do n't get me wrong , its no pumpking , but this is a damn fine pumpkin ale that could hold its own any day among all the others . i would rate this as my 4th favorite pumpkin ale to date . i 'm not sure why the bros rated it so low , but do n't take their opinion , make your own !

Figure 5: MAM can identify accurately what parts of the review describe each aspect. Due to the high imbalance and correlation, MAA provides very noisy labels, while MASA highlights only a few important words. We can see that SAM is confused and performs a poor selection.

Figure 6: On a short review, MAM achieves near-perfect annotations, while SAM highlights only two words where one is ambiguous with respect to four aspects. MAA mixes between the aspect *Appearance* and *Smell*. MASA identifies some words but lacks coverage.

### A.3.3 HOTEL DATASET

Service Cleanliness Value Location Room

**Multi Aspect-Masks (Ours)**

Service Cleanliness Value Location Room

Single Aspect-Mask (Lei et al., 2016)

i stayed at daulsol in september 2013 and could n't have asked for anymore for the price ! ! it is a great location .... only 2 minutes walk to jet , space and sankeys with a short drive to ushuaia . the hotel is basic but cleaned daily and i did nt have any problems at all with the bathroom or kitchen facilities . the lady at reception was really helpful and explained everything we needed to know ..... even when we managed to miss our flight she let us stay around and use the facilities until we got on a later flight . there are loads of restaurants in the vicinity and supermarkets and shops right outside . i loved these apartments so much that i booked to come back for september 2014 ! ! can not wait :)

i stayed at daulsol in september 2013 and could n't have asked for anymore for the price ! ! it is a great location .... only 2 minutes walk to jet , space and sankeys with a short drive to ushuaia . the hotel is basic but cleaned daily and i did nt have any problems at all with the bathroom or kitchen facilities . the lady at reception was really helpful and explained everything we needed to know ..... even when we managed to miss our flight she let us stay around and use the facilities until we got on a later flight . there are loads of restaurants in the vicinity and supermarkets and shops right outside . i loved these apartments so much that i booked to come back for september 2014 ! ! can not wait :)

Service Cleanliness Value Location Room

Multi Aspect-Attentions

Service Cleanliness Value Location Room

Multi Aspect-Sparse-Attentions

i stayed at daulsol in september 2013 and could n't have asked for anymore for the price ! ! it is a great location .... only 2 minutes walk to jet , space and sankeys with a short drive to ushuaia . the hotel is basic but cleaned daily and i did nt have any problems at all with the bathroom or kitchen facilities . the lady at reception was really helpful and explained everything we needed to know ..... even when we managed to miss our flight she let us stay around and use the facilities until we got on a later flight . there are loads of restaurants in the vicinity and supermarkets and shops right outside . i loved these apartments so much that i booked to come back for september 2014 ! ! can not wait :)

i stayed at daulsol in september 2013 and could n't have asked for anymore for the price ! ! it is a great location .... only 2 minutes walk to jet , space and sankeys with a short drive to ushuaia . the hotel is basic but cleaned daily and i did nt have any problems at all with the bathroom or kitchen facilities . the lady at reception was really helpful and explained everything we needed to know ..... even when we managed to miss our flight she let us stay around and use the facilities until we got on a later flight . there are loads of restaurants in the vicinity and supermarkets and shops right outside . i loved these apartments so much that i booked to come back for september 2014 ! ! can not wait :)

Figure 7: MAM emphasizes consecutive words, identifies important spans while having a small amount of noise. SAM focuses on certain specific words and spans, but labels are ambiguous. The MAA model highlights many words, ignores a few important key-phrases, and labels are noisy when the confidence is not high. MASA provides noisier tags than MAA.

Service Cleanliness Value Location Room

**Multi-Aspect Masker (Ours)**

stayed at the parasio 10 apartments early april 2011 . reception staff absolutely fantastic , great customer service .. ca nt fault at all ! we were on the 4th floor , facing the front of the hotel .. basic , but nice and clean , good location , not too far away from the strip and beach ( 10 min walk ) . however .. do not go out alone at night at all ! i went to the end of the street one night and got mugged .. all my money , camera .. everything ! got sratches on my chest which has now scarred me , and i had bruises at the time . just make sure you have got someone with you at all times , the local people are very renound for this . went to police station the next day ( in old town ) and there was many english in there reporting their muggings from the day before . shocking ! ! apart from this incident ( on the first night we arrived :( ) we had a good time in the end , plenty of laughs and everything is very cheap ! beer - 1euro ! fryups - 2euro . would go back again , but maybe stay somewhere else closer to the beach ( sol pelicanos etc ) .. this hotel is next to an alley called ' muggers alley '

Service Cleanliness Value Location Room

Single Aspect-Mask (Lei et al., 2016)

stayed at the parasio 10 apartments early april 2011 . reception staff absolutely fantastic , great customer service , ca nt fault at all ! we were on the 4th floor , facing the front of the hotel .. basic , but nice and clean . good location , not too far away from the strip and beach ( 10 min walk ) . however .. do not go out alone at night at all ! i went to the end of the street one night and got mugged .. all my money , camera .. everything ! got sratches on my chest which has now scarred me , and i had bruises at the time . just make sure you have got someone with you at all times , the local people are very renound for this . went to police station the next day ( in old town ) and there was many english in there reporting their muggings from the day before . shocking ! ! apart from this incident ( on the first night we arrived :( ) we had a good time in the end , plenty of laughs and everything is very cheap ! beer - 1euro ! fryups - 2euro . would go back again , but maybe stay somewhere else closer to the beach ( sol pelicanos etc ) .. this hotel is next to an alley called ' muggers alley '

Service Cleanliness Value Location Room

Multi Aspect-Attentions

stayed at the parasio 10 apartments early april 2011 . reception staff absolutely fantastic , great customer service .. ca nt fault at all ! we were on the 4th floor , facing the front of the hotel .. basic , but nice and clean . good location , not too far away from the strip and beach ( 10 min walk ) . however .. do not go out alone at night at all ! i went to the end of the street one night and got mugged .. all my money , camera .. everything ! got sratches on my chest which has now scarred me , and i had bruises at the time . just make sure you have got someone with you at all times , the local people are very renound for this . went to police station the next day ( in old town ) and there was many english in there reporting their muggings from the day before . shocking ! ! apart from this incident ( on the first night we arrived :( ) we had a good time in the end , plenty of laughs and everything is very cheap ! beer - 1euro ! fryups - 2euro . would go back again , but maybe stay somewhere else closer to the beach ( sol pelicanos etc ) .. this hotel is next to an alley called ' muggers alley '

Service Cleanliness Value Location Room

Multi Aspect-Sparse-Attentions

stayed at the parasio 10 apartments early april 2011 . reception staff absolutely fantastic , great customer service .. ca nt fault at all ! we were on the 4th floor , facing the front of the hotel .. basic , but nice and clean . good location , not too far away from the strip and beach ( 10 min walk ) . however .. do not go out alone at night at all ! i went to the end of the street one night and got mugged .. all my money , camera .. everything ! got sratches on my chest which has now scarred me , and i had bruises at the time . just make sure you have got someone with you at all times , the local people are very renound for this . went to police station the next day ( in old town ) and there was many english in there reporting their muggings from the day before . shocking ! ! apart from this incident ( on the first night we arrived :( ) we had a good time in the end , plenty of laughs and everything is very cheap ! beer - 1euro ! fryups - 2euro . would go back again , but maybe stay somewhere else closer to the beach ( sol pelicanos etc ) .. this hotel is next to an alley called ' muggers alley '

Figure 8: Our MAM model finds most of the important span of words with a small amount of noise. SAM lacks coverage but identifies words where half are correctly tags and the others ambiguous. MAA partially correctly highlights words for the aspects *Service*, *Location*, and *Value* while missing out the aspect *Cleanliness*. MASA confidently finds a few important words.

## A.4  TOPIC WORDS PER ASPECT

For each model, we computed the probability distribution of words per aspect by using the induced sub-masks $M_{a_1}, ..., M_{a_A}$ or attention values. Given an aspect $a_i$ and a set of top-$N$ words $\boldsymbol{w_{a_i}^N}$, the Normalized Pointwise Mutual Information (Bouma, 2009) coherence score is:

$$\text{NPMI}(\boldsymbol{w_{a_i}^N}) = \sum_{j=2}^{N}\sum_{k=1}^{j-1} \frac{\log \frac{P(w_{a_i}^k, w_{a_i}^j)}{P(w_{a_i}^k)P(w_{a_i}^j)}}{-\log P(w_{a_i}^k, w_{a_i}^j)} \tag{6}$$

Top words of coherent topics (i.e., aspects) should share a similar semantic interpretation and thus, interpretability of a topic can be estimated by measuring how many words are not related. For each aspect $a_i$ and word $w$ having been highlighted at least once as belonging to aspect $a_i$, we computed the probability $P(w|a_i)$ on each dataset and sorted them in decreasing order of $P(w|a_i)$. Unsurprisingly, we found that the most common words are stop words such as "a" and "it", because masks are mostly word sequences instead of individual words. To gain a better interpretation of the aspect words, we followed the procedure in McAuley et al. (2012): we first computed averages across all aspect words for each word $w$: $b_w = \frac{1}{|A|}\sum_{i=1}^{|A|} P(w|a_i)$, which generates a general distribution that includes words common to all aspects. The final word distribution per aspect is computed by removing the general distribution: $\hat{P}(w|a_i) = P(w|a_i) - b_w$.

After generating the final word distribution per aspect, we picked the top ten words and asked two human annotators to identify intruder words, i.e., words not matching the corresponding aspect. We show in subsequent tables the top ten words for each aspect, where **red** denotes all words identified as unrelated to the aspect by the two annotators. Generally, our model finds better sets of words across the three datasets compared with other methods. Additionally, we observe that the aspects can be easily recovered given its top words.

Table 7: Top ten words for each aspect from the ***Filtered Beer*** dataset, learned by various models. **Red** denotes intruders according to two human annotators. For the three aspects, MAM has only one word considered as an intruder, followed by MASA with SAM (two) and MAA (six).

| | Model | Top-10 Words |
|---|---|---|
| *Appearance* | SAM | head color white brown dark lacing **pours** amber clear black |
| | MASA | head lacing lace retention glass foam color amber yellow cloudy |
| | MAA | nice dark amber **pours** black hazy brown **great** cloudy clear |
| | MAM (Ours) | head color lacing white brown clear amber glass black retention |
| *Smell* | SAM | sweet malt hops coffee chocolate citrus hop strong smell aroma |
| | MASA | smell aroma nose smells sweet aromas scent hops malty roasted |
| | MAA | **taste** smell aroma sweet chocolate **lacing** malt roasted hops nose |
| | MAM (Ours) | smell aroma nose smells sweet malt citrus chocolate caramel aromas |
| *Palate* | SAM | mouthfeel smooth medium carbonation bodied watery body thin creamy **full** |
| | MASA | mouthfeel medium smooth body **nice** m- feel bodied mouth **beer** |
| | MAA | carbonation mouthfeel medium **overall** smooth finish body **drinkability** bodied watery |
| | MAM (Ours) | mouthfeel carbonation medium smooth body bodied **drinkability** good mouth thin |

Table 8: Top ten words for each aspect from the ***Full Beer*** dataset, learned by various models. **Red** denotes intruders according to two annotators. Found words are generally noisier due to the high correlation between *Taste* and other aspects. However, MAM provides better results than other methods.

| | Model | Top-10 Words |
|---|---|---|
| *Apperance* | SAM | **nothing** beautiful lager nice **average** macro lagers corn **rich** gorgeous |
| | MASA | lacing head lace **smell** amber retention beer nice carbonation glass |
| | MAA | head lacing **smell aroma** color pours amber glass white retention |
| | MAM (Ours) | head lacing **smell** white lace retention glass **aroma** tan thin |
| *Smell* | SAM | faint **nice mild** light slight complex good wonderful grainy great |
| | MASA | aroma hops nose chocolate caramel malt citrus fruit smell fruits |
| | MAA | **taste** hints hint **lots t- starts** blend mix **upfront** malts |
| | MAM (Ours) | **taste** malt aroma hops sweet citrus caramel nose malts chocolate |
| *Palate* | SAM | thin **bad** light watery creamy silky medium body smooth **perfect** |
| | MASA | smooth light medium thin creamy **bad** watery **full** crisp **clean** |
| | MAA | good beer carbonation smooth **drinkable** medium bodied **nice** body **overall** |
| | MAM (Ours) | carbonation medium mouthfeel body smooth bodied **drinkability** creamy light **overall** |
| *Taste* | SAM | **decent** great complex delicious tasty favorite **pretty** sweet **well best** |
| | MASA | good **drinkable nice** tasty great enjoyable **decent solid** balanced **average** |
| | MAA | malt hops flavor hop flavors caramel malts bitterness bit chocolate |
| | MAM (Ours) | malt sweet hops flavor bitterness finish chocolate bitter caramel sweetness |

Table 9: Top ten words for each aspect from the ***Hotel*** dataset, learned by various models. **Red** denotes intruders according to human annotators. Besides SAM, all methods find similar words for most aspects except the aspect *Value*, where MAM does not have an intruder.

| | Model | Top-10 Words |
|---|---|---|
| *Service* | SAM | staff service friendly nice told helpful good great lovely manager |
| | MASA | friendly helpful told rude nice good pleasant asked enjoyed worst |
| | MAA | staff service helpful friendly nice good rude excellent great desk |
| | MAM (Ours) | staff friendly service desk helpful manager reception told rude asked |
| *Cleanliness* | SAM | clean cleaned dirty toilet smell cleaning sheets comfortable nice hair |
| | MASA | clean dirty cleaning spotless stains cleaned cleanliness mold filthy bugs |
| | MAA | clean dirty cleaned filthy stained well spotless carpet sheets stains |
| | MAM (Ours) | clean dirty bathroom room bed cleaned sheets smell carpet toilet |
| *Value* | SAM | good stay great well **dirty** recommend worth definitely **friendly** charged |
| | MASA | great good poor excellent terrible awful **dirty** horrible **disgusting comfortable** |
| | MAA | **night stayed** stay **nights 2 day** price **water 4 3** |
| | MAM (Ours) | good price expensive paid cheap worth better pay overall disappointed |
| *Location* | SAM | location close far place walking **definitely** located **stay** short view |
| | MASA | location beach walk hotel town located restaurants walking close taxi |
| | MAA | location hotel place located close far area beach view situated |
| | MAM (Ours) | location great area walk beach hotel town close city street |
| *Room* | SAM | **dirty clean** small best comfortable large worst modern **smell** spacious |
| | MASA | comfortable small spacious nice large dated well tiny modern basic |
| | MAA | room rooms bathroom bed spacious small beds large shower modern |
| | MAM (Ours) | comfortable room small spacious nice modern rooms large tiny walls |

## A.5 BASELINE ARCHITECTURES

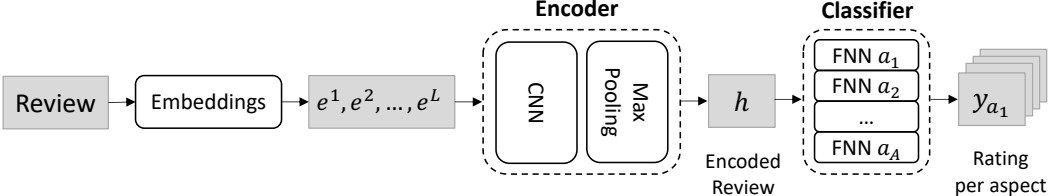

Figure 9: Baseline model Emb + Enc$_{\text{CNN}}$ + Clf.

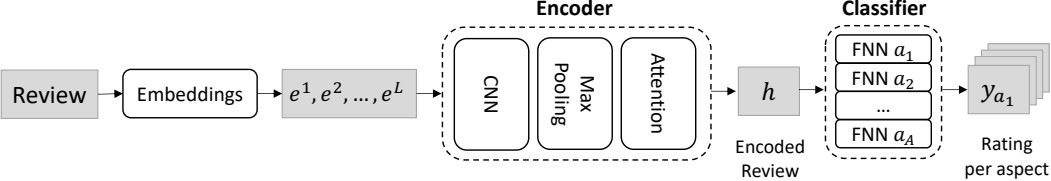

Figure 10: Baseline model Emb + Enc$_{\text{CNN}}$ + A$_{\text{Shared}}$ + Clf.

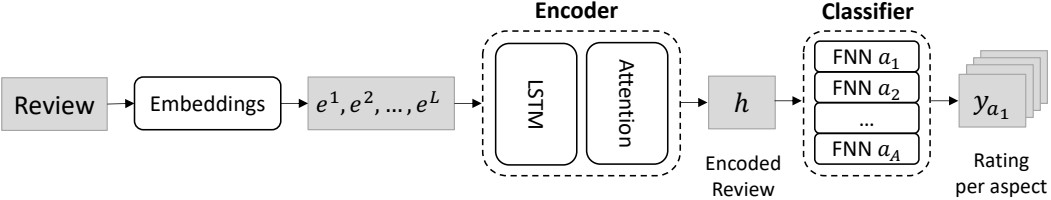

Figure 11: Baseline model Emb + Enc$_{\text{LSTM}}$ + A$_{\text{Shared}}$ + Clf.

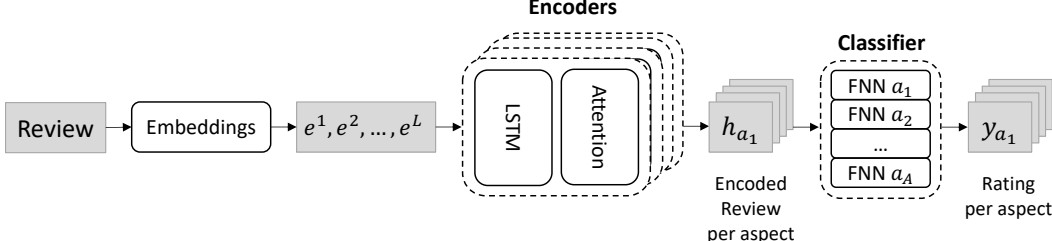

Figure 12: Baseline model Emb + Enc$_{\text{LSTM}}$ + A$^{[\text{Sparse}]}_{\text{Aspect-wise}}$ + Clf. Attention is either additive or sparse.

