# OpenReview forum: "Multi-Dimensional Explanation of Reviews"
_ICLR.cc/2020/Conference — Reject_

### Official Review · AnonReviewer1 · 2019-10-06
**Official Blind Review #1**

**Rating:** 3

**Review:**

The paper is mostly concerned with multi-aspect sentiment classification.

Multi-aspect sentiment analysis is an old problem (> 10 years), that usually operates on datasets with multidimensional sentiment labels (price, cleanliness, etc.). This paper has some variation on this old problem in the sense that it's more focused on generating explanations / justifications rather than purely on sentiment analysis.

Although an interesting variant on a (somewhat niche) existing problem, I do feel that some aspects of this paper are a little dated -- claiming that the method is more useful for interpretability / justification doesn't seem so convincing these days unless you can make a stronger argument as to how it could actually be used to generate justifications that would be shown to users. I don't really see that in the current version. At the moment the output of the system appears to be attention weights over several aspects, showing which words in a review focus on which aspect. But this doesn't seem so different from how older methods (e.g. Bag-of-Words based techniques for multi-aspect sentiment classification), though their labels were possibly at the level of sentences rather than reviews.

The actual method itself seems appropriate, and is a nice update on previous techniques for multi-aspect sentiment analysis.

Perhaps I missed some detail explaining why such a comparison is impossible, but there seems to be no direct comparison against traditional techniques for multi-aspect sentiment analysis. E.g. aren't even traditional models (like say Titov & McDonald) comparable to this? I understand the goal is somewhat different, but the input/output modality seems similar and such a comparison could be helpful.

Overall, I think these days the standard is quite high for papers that make claims about generating convincing justifications/interpretations/explanations. There's little by way of human evaluation (there's a bit in the appendix, but it's not really focused on whether or not the model is useful for the claimed purpose of "explanation").

More positively, the authors have done *a lot* by way of experiments, in terms of ablations, and showing quantitative performance from different dimensions. The appendix is detailed and shows many examples. Overall I feel somewhat borderline about this paper, because there's a lot here, but I feel some of these key components mentioned above are missing.

**Experience Assessment:**

I have published one or two papers in this area.

**Review Assessment: Checking Correctness Of Derivations And Theory:**

I did not assess the derivations or theory.

**Review Assessment: Checking Correctness Of Experiments:**

I assessed the sensibility of the experiments.

**Review Assessment: Thoroughness In Paper Reading:**

I made a quick assessment of this paper.

---

> ### Author Response · Authors · 2019-11-11
> **Our model generates explanations at the word-level and that are beneficial for users or models**
>
> Thank you for your feedback.
>
> First of all, we would like to highlight that our paper is about providing interpretable fragments of texts to justify the predicted sentiment towards an aspect, in a multi-aspect sentiment analysis task, as opposed to simply predicting the sentiments. We show that our model provides meaningful interpretable masks while achieving better performance than other models.
>
> We would like to answer to your comments:
>
> -	“claiming that the method is more useful for interpretability / justification doesn't seem so convincing these days unless you can make a stronger argument as to how it could actually be used to generate justifications that would be shown to users … But this doesn't seem so different from how older methods (e.g. Bag-of-Words based techniques for multi-aspect sentiment classification), though their labels were possibly at the level of sentences rather than reviews.”
> As shown in Figure 2 and Appendices A.3.1-3, the end goal of the method is to provide interpretability towards each aspect and either show it to a user (for example while a user reads reviews of an hotel, he or she could directly see what part of the reviews talk about what aspect) or use generated masks in other tasks such as recommender system, review summarization etc.
> Moreover, our method is different as it predicts the sentiment towards an aspect and highlights one or several sequences of words justifying the prediction (see Section 2).
>
> -	“Perhaps I missed some detail explaining why such a comparison is impossible, but there seems to be no direct comparison against traditional techniques for multi-aspect sentiment analysis.”
> Our method is different than traditional methods because it provides interpretability in addition to sentiments towards each aspect, as highlighted in the Introduction, Section 2.3 and 3.1.
>
> -	“There's little by way of human evaluation (there's a bit in the appendix, but it's not really focused on whether or not the model is useful for the claimed purpose of "explanation").” “More positively, the authors have done *a lot* by way of experiments, in terms of ablations, and showing quantitative performance from different dimensions. The appendix is detailed and shows many examples. Overall I feel somewhat borderline about this paper, because there's a lot here, but I feel some of these key components mentioned above are missing.”
> We quantitively evaluate interpretability in Table 4 (on an annotated dataset) and Table 5 (topic model metrics). We qualitatively show the interpretability of our masks in Figure 2 and Appendices A.3.1-3 on 6 samples. If you have suggestions for a better evaluation, we would be happy to consider it.

---

### Official Review · AnonReviewer3 · 2019-10-23
**Official Blind Review #3**

**Rating:** 3

**Review:**

This paper proposes a neural model for multi-aspect sentiment of reviews. Apart from sentiment classification, the architecture is able to return a mask for each aspect highlighting the sequences of words used to make the sentiment prediction for that aspect. This improves the interpretability of the model. The aspects are fixed and the training set requires a multi-labeled dataset with one binary positive/negative label per aspect.

The network architecture is mostly standard: a text review is turned into a list of word embeddings which are fed through a trainable masker outputting one mask per aspect, essentially "selecting" a different set of words (embeddings) per aspect. From here on, a standard encoder can be used (the authors end up using a CNN with max-pooling but LSTM or other could be used), followed by a classification step (standard MLP) outputting a sentiment score per aspect. A couple tricks are used to make the masks converge to something meaningful by adding regularization terms to control the number of selected words and encouraging consecutive words to be selected.

I'm leaning toward rejection partly due to the limited novelty in the approach but mainly due to the experimental evaluation section which requires improvements.
- High aspect correlation baseline: the average aspect correlation across reviews in the training set is high (that is, if one aspect is positive, most aspects will be and viceversa). A natural baseline to include is the optimal "no-aspect" decision boundary: in other words, if for each review we were given the majority rating across aspects (is this review mostly positive or negative) and were able to predict this always correctly, what would the macro-aspect F1 score be if we were to simply predict that label to all aspect of that review?
- Please quantify how much does your model rely on this aspect correlation. For example, if you were to sort the reviews in the validation set by their aspect correlation, and chart the F1 score as a function of the prefix of this sorted dataset what would we see? This would be especially telling when comparing the different baselines and classifiers again each other in this chart.
- SVM: you mentioned not running it on the larger datasets due to lengthy training time. I would suggest including NB-SVM implemented with logistic regression rather than actual SVM since it's always a great baseline and fast to compute (in "Baselines and Bigrams: Simple, Good Sentiment and Topic Classification").
- The final model that ends up giving the best results is really a 2-step pipeline (rather than end-to-end trainable) and requires training twice: once to learn the masks and once to actually train the rest of the network after the word embeddings are extended with the masks.













**Experience Assessment:**

I have read many papers in this area.

**Review Assessment: Checking Correctness Of Derivations And Theory:**

N/A

**Review Assessment: Checking Correctness Of Experiments:**

I assessed the sensibility of the experiments.

**Review Assessment: Thoroughness In Paper Reading:**

I read the paper at least twice and used my best judgement in assessing the paper.

---

> ### Author Response · Authors · 2019-11-11
> **Added Majority-Sentiment, NB-SVM and addressed novelty**
>
> Thank you very much for your feedback. We were able to incorporate your suggestions and believe it has strengthened our paper.
>
> We have updated our results by including the “Majority Sentiment” baseline you proposed and using NB-SVM on the three datasets. Regarding your last remark, our model MAM provides better interpretability with a single training (see Section 4.4 and 4.5) and outperforms other models on two out of three datasets and is competitive on the Hotel dataset. We also show that by using the mask as additional features on a smaller model, we obtain the best results across all the datasets, while providing the same interpretability.
>
> We would like to insist on the novelty of our approach. As pointed out in Section 3 (the paragraph above Equation 2) and Section 2.2, the novelty of the method is 1) the use of two regularizers to control the number of selected words and encourage consecutive words to be selected, and 2) the normalization not done over the sequence length but over the aspects and non-aspect set (the Masker component).
> As a consequence, across three datasets, our model outperforms strong baselines and generates masks that are: strong feature predictors, meaningful, and interpretable. The Figure 2 and Appendices A.3.1-3 qualitatively show that our model highlights texts that are more meaningful. In addition, Table 4 quantitively underlines that MAM obtained significantly higher precision of selected words for each aspect, on an annotated dataset; Table 5 shows similar trends with topic model metrics.
>
> Regarding your comment about the aspect correlation, we find the idea very interesting. However, we don’t see clearly how you would compute the aspect correlation for a single review as we could do it only on the whole dataset, as shown in Table 6 in Appendix A.1. Do you have any suggestion?

---

### Official Review · AnonReviewer2 · 2019-10-23
**Official Blind Review #2**

**Rating:** 6

**Review:**

This work addresses the problem of aspect-based sentiment analysis *with* rationale extraction that can serve as an explanation (interpretation) of the decision.
Compared to previous work this paper models the problem as multi-aspect classification, therefore having one model for all aspects instead of one for each aspect. This is obviously useful, but in addition to having a smaller total model they show that they also have (slightly) better prediction AND AT THE SAME TIME "better" explanations (which is always hard to evaluate).

This is a useful paper, and I can clearly see it used. However, it might not get many ICLR attendants excited.

I found the paper very hard to read, having to re-read many parts to understand exactly what is the goal of each section. The row names of the table 1-3 do not correspond to the names in the text which makes the understanding even harder. Also, I think that some plot that shows accuracy vs "interpretability", with model size as a third dimension (eg surface of the bubble) would convey the take-home of the paper much clearer than the many tables with small differences between them.

A minor comment: the SVM has a huge parameter space, which is weird. Are you using *all* bigrams? A fairer comparison would be to use more n-grams, but keep only the most frequent ones.

**Experience Assessment:**

I have read many papers in this area.

**Review Assessment: Checking Correctness Of Derivations And Theory:**

I assessed the sensibility of the derivations and theory.

**Review Assessment: Checking Correctness Of Experiments:**

I assessed the sensibility of the experiments.

**Review Assessment: Thoroughness In Paper Reading:**

I read the paper at least twice and used my best judgement in assessing the paper.

---

> ### Author Response · Authors · 2019-11-11
> **Improved readability, modified SVM**
>
> Thank you for your valuable feedback, which we have incorporated.
>
> We have improved the readability of the tables by incorporating a row-number for each model and referring to them within the text of Section 4.3.
>
> Regarding your comment about SVM, we have decreased the number of bigrams used, and employed the NB-SVM model of Wang & Manning (2012), as suggested by Reviewer#3. Therefore, we could train the model on the Full Beer dataset and have updated the results on the two other datasets.
>
> Thank you for the proposition concerning the visualization. We thought about it and found that it could work if we report only the Macro F1 Score instead of the Macro & individual F1 Scores. If you have a suggestion on how to best display all data, we would be happy to hear it.

---

### Author Response · Authors · 2019-11-11
**Summary of changes, Nov 11, 2019.**

We thank all three reviewers for their valuable and constructive feedback, which we have used to improve the paper. We summarize the changes here and individually replied to each reviewer below.

We changed our baseline SVM with the model NB-SVM of Wang & Manning (2012). Consequently, we could train it on the Full Beer dataset and decrease the number of parameters to fairly compare with other models.

As the sentiment correlation between any pair of aspects might be high, we included a new baseline “Majority Sentiment,” which for a review outputs the majority sentiment across aspects.

Finally, we incorporated a row-number for each model in Table 1-3 to improve the readability of Section 4.3.

---

### Decision · Program_Chairs · 2019-12-19

**Decision:**

Reject

**Comment:**

This paper proposes a neural network model for predicting multi-aspect sentiment and generating masks that can justify the predictions. The positive aspects of the paper include improved results over the state-of-the-art.

Reviewers found the technical novelty limited, and the experiments short of being fully convincing. After the author rebuttal, there were discussions between the reviewers and the AC, and the reviewers still thought the paper is not fully convincing given these limitations.

I thank the authors for their submission and detailed responses to the reviewers and hope to see this research in a future venue.